# The Emerging Role of Peroxisome Proliferator-Activated Receptors in Cancer Stemness

**DOI:** 10.3390/cells14201610

**Published:** 2025-10-16

**Authors:** Beatriz Parejo-Alonso, Marta Mascaraque, Alba Royo-García, Patricia Sancho

**Affiliations:** 1Instituto de Investigación Sanitaria Aragón (IIS Aragón), 50009 Zaragoza, Spain; bparejo@iisaragon.es (B.P.-A.); marta.mascaraque@uam.es (M.M.); aroyo@iisaragon.es (A.R.-G.); 2Department of Biology, Universidad Autónoma de Madrid, 28049 Madrid, Spain

**Keywords:** PPARs, cancer stem cells, self-renewal, chemoresistance, metastasis, immune evasion

## Abstract

**Highlights:**

**What are the main findings?**
PPARs regulate cancer aggressiveness and stemness.PPAR functions in cancer are tissue- and context-dependent.

**What is the implication of the main finding?**
PPARs represent potential therapeutic targets for CSC targeting.Pleiotropic effects of PPAR modulation may hinder their efficiency.

**Abstract:**

The peroxisome proliferator-activated receptors (PPAR-α, PPAR-δ, and PPAR-γ) are transcription factors that belong to the nuclear hormone receptor superfamily. Upon activation by specific lipids, they regulate gene expression by directly binding to PPAR response elements (PPREs) in the DNA. Although the functions of the different PPARs are specific to the isoform, tissue, and context, all three PPARs are generally involved in energy homeostasis through lipid sensing in physiological conditions. Importantly, there is increasing evidence linking PPARs with malignant behavior in cancer, regulating features frequently attributed to the aggressive subpopulation of cancer stem cells (CSCs): self-renewal, tumor initiation, chemoresistance, metastasis, and immune evasion. However, contradictory effects have been described for each isoform in various cancer types, and their implication in these malignant features may not consistently follow a pro- or anti-tumoral pattern. In this review, we revise the current knowledge on the role of the PPAR family members in cancer, with a special focus on cancer stemness, and discuss the potential for PPARs as therapeutic targets in CSC-driven relapse and resistance.

## 1. Peroxisome Proliferator-Activated Receptors

The peroxisome proliferator-activated receptors (PPARs) comprise a family of proteins belonging to the nuclear hormone receptor superfamily of transcription factors (TFs) that regulate gene expression by directly binding to PPAR response elements (PPREs) in the DNA, upon activation by specific ligands [1]. Three different PPAR isoforms have been described to date: PPAR-α (NR1C1), PPAR-δ (also known as PPAR-β or NR1C2), and PPAR-γ (NR1C3). The PPAR structure consists of four main regions: (1) A/B or ligand-independent domain, which includes the AF1 region, important for subtype-specific activity; (2) C or DNA-binding domain (DBD); (3) D or docking cofactors domain; and (4) E or ligand-binding domain (LBD). The PPAR LBD domain is larger than in other nuclear receptors, facilitating the interaction with a wider range of compounds, all of them harboring an acidic head group [2]. Moreover, while PPAR-α and PPAR-γ have a similar ligand-binding pocket in terms of size and shape, that of PPAR-δ is significantly smaller, explaining the observed ligand restriction in PPAR-δ versus the other two isoforms. The activity of the different PPARs can be modulated by endogenous ligands such as fatty acids (FAs) and FA-derived molecules or synthetic compounds [1]. Upon ligand-dependent activation, all PPARs heterodimerize with the retinoid X receptor (RXR), which belongs to the same receptor superfamily, and the LBD acquires a conformation that allows the binding of coactivator or corepressor proteins. Then, a complex formed by the PPAR:RXR heterodimer plus the coactivator or corepressor binds to the PPREs present in the promoters of the target genes, activating or repressing their expression [3]. Although the function of the different PPARs is isotype-specific and tissue- and context-dependent, in general all three PPARs are involved in energy homeostasis through lipid sensing, while PPAR-δ additionally regulates cell proliferation, differentiation, survival, and tissue repair (Figure 1). Importantly, mounting evidence associates the PPARs with tumorigenesis, rendering them excellent candidates for therapeutic intervention with the different targeted pharmacological strategies currently available [1,4].

### 1.1. PPAR-Alpha

PPAR-α primarily acts as a nutritional sensor and controls the expression of genes involved in lipid transport and fatty acid oxidation (FAO) to maintain lipid homeostasis, which is essential for overall bodily function [5]. It is predominantly expressed in tissues with high FA catabolism rates, such as the liver, heart, kidney, and muscle. For instance, it plays a crucial role in maintaining lipid homeostasis in the liver, where it regulates apolipoprotein expression, enhances FAO, reduces triglyceride levels, and promotes ketogenesis. In the muscle, PPAR-α facilitates the use of FAs as an energy source.

The expression of *PPARA* is influenced by lipidic metabolites. Indeed, saturated and monounsaturated FAs, as well as docosahexaenoic acid (DHA), increase *PPARA* mRNA expression, while linolenic and eicosapentaenoic acids decrease it [6,7]. Knockout studies of liver-specific fatty acid synthase (*FAS*) have shown that PPAR-α agonists can reverse non-alcoholic steatohepatitis, indicating that FAS products act as PPAR-α activators [8]. Furthermore, it has been shown that disruption of ACOX1 (peroxisomal acyl-coenzyme A oxidase 1), the first enzyme of FAO, led to elevated expression of PPAR-α target genes, suggesting that ACOX1 substrates also function as PPAR-α agonists [9]. Finally, hydrolysis of triglycerides also produces PPAR-α ligands [10].

PPAR-α regulates FA uptake by controlling the expression of FA transporters, such as *FATP1*, *SLC27A1*, and *FAT*/*CD36* [11,12,13]. In addition, it controls FAO in mitochondria and peroxisomes by directly regulating the expression of mitochondrial acyl-CoA dehydrogenases (e.g., *VLCAD*, *LCAD*, and *MCAD*) [14] or peroxisomal enzymes (e.g., *ACOX1* and *EHHADH*), respectively [15]. *PPARA* expression can also be upregulated by the metabolic stress sensor AMPK, which then induces the expression of PPAR-α target genes, such as *CPT1* and *PGC-1*, and increases FAO in skeletal muscle [16,17].

PPAR-α may play a dual role in cancer, demonstrating both protumorigenic and anti-tumorigenic effects depending on the cellular context and tumor type. As mentioned earlier, PPAR-α regulates lipid metabolism, FA homeostasis, and oxidative stress, processes that are crucial for cancer development and progression [6,13,18]. PPAR-α has also been shown to play a critical role in the regulation of genes involved in the initiation and progression of hormone-dependent tumors, including breast cancer, through the biotransformation of endogenous estrogens and environmental carcinogens [19]. Elevated *PPARA* expression has been demonstrated in several cancers, such as ampullary cancer [20] and stomach adenocarcinomas [21].

On the other hand, PPAR-α exhibits anticancer effects. The repression of *PPARA* mediated by protein tyrosine phosphatase receptor type O in colon cancer has been associated with a worse prognosis [22]. Moreover, loss of PPAR-α in the intestine promoted colon carcinogenesis by increasing DNA methyltransferase 1-mediated methylation of P21 and protein arginine methyltransferase 6-mediated methylation of P27 in mice [23]. Similar effects have been observed in breast cancer, where PPAR-α agonists enhanced the efficacy of immunotherapy [24] and chemotherapy [25,26], while mitigating side effects.

### 1.2. PPAR-Delta

PPAR-*δ* exerts a wide range of physiological functions in a tissue-specific manner, most of which are related to lipid metabolism. It is expressed broadly, especially in organs with high FA metabolism, reproductive organs, and the cardiovascular, endocrine, and immune systems, with increased basal levels in the gastrointestinal tract and skeletal muscle [2].

Unsaturated FAs, including derivatives of linoleic acid, are natural ligands for PPAR-*δ*. However, the effect of endogenous ligand-dependent activation of PPAR-*δ* is, in fact, tissue-specific, also depending on the relative presence of coactivators and corepressors. For instance, 13-HODE (13-hydroxyoctadecadienoic acid) has been reported to inhibit PPAR-*δ* in colorectal cancer cells to induce apoptosis [27], while in pre-adipocytes, it acts as an agonist, enhancing lipid detoxification [28]. Additionally, PPAR-*δ* may be exogenously activated with synthetic ligands. Interestingly, PPAR-*δ* activity can be positively regulated via phosphorylation by protein kinase A (PKA) or inhibited by ubiquitin-induced proteolysis [29].

The main function of PPAR-*δ* involves the regulation of the balance between glucose and lipid metabolism. In the pancreas, *PPARD* knockout increased insulin secretion, while systemic treatment of obese (ob/ob) [30] and diabetic (db/db) [31] mice with PPAR-*δ* agonists enhanced glucose-stimulated insulin secretion and normalized pancreatic islet hypertrophy. In the liver, one of the most important organs implicated in energy homeostasis, PPAR-*δ* promotes lipogenesis and activates the pentose phosphate pathway (PPP) to increase glucose utilization, while *Ppard* null mice exhibited glucose intolerance [32].

The role of PPAR-*δ* in cancer is certainly controversial, as both pro- and anti-tumoral functions have been reported in different cancer types. For instance, while ligand-independent activation of PPAR-*δ* exerted tumor suppressive functions in prostate cancer by repressing trefoil factor 1 (*TFF1*) [33], *PPARD* expression has been correlated with cancer progression, angiogenesis, and metastasis in several cancers [34,35]. In liver cancer, *Ppard^−/−^* mice were more susceptible to developing induced hepatocellular carcinoma (HCC) and, accordingly, *PPARD* overexpression in HCC cell lines inhibited proliferation, migration, and invasion while enhancing apoptosis in these cells [36]. A different study correlated increased PPAR-*δ* activity to treatment resistance in HCC [37]. Importantly, contradictory roles have been described for PPAR-*δ* even within the same malignancy, probably explained by the different experimental models used [38]. For instance, *PPARD* knockdown promoted proliferation and resistance to treatment in vitro in the established colon cancer cell line KM12C [39]. Contrastingly, PPAR-*δ* activation endowed intestinal organoid- and tumor-initiating properties induced by a high-fat diet (HFD) or treatment with synthetic ligands in colon cancer [40]. Contradictory results have also been published for breast cancer cell lines, reporting both PPAR-*δ*-dependent inhibition of proliferation and tumorigenicity [41], and proliferative, survival, and metastatic advantages in normal and harsh metabolic conditions [42]. In these reports, a similar overexpression system in MCF7 cells and other established cell lines was used, so differences could be attributed to experimental design. Likewise, PPAR-*δ* was initially found to inhibit invasion in MiaPaca and BXPC3 pancreatic cancer cells [43] in response to TNF-α in vitro, but different studies revealed its implication in epithelial-to-mesenchymal transition (EMT), migration, and invasion, as well as in vivo metastasis, in pancreatic and other cancers [34,44]. Interestingly, recent studies demonstrate a protumoral role for PPAR-δ by creating an immunosuppressive microenvironment via transcriptional regulation of cytokines and chemokines that altered the recruitment of immune cells [45,46].

### 1.3. PPAR-Gamma

PPAR-*γ* is recognized as the master regulator of adipose tissue biology. The gene *PPARG* encodes four isoforms, with PPAR-*γ*1 and PPAR-*γ*2 being the most studied. *PPARG1* is expressed in a number of tissues, including adipose tissue, liver, colon, heart, skeletal muscle, and various immune cells such as monocytes/macrophages, dendritic cells, and T lymphocytes [47]. In contrast, *PPARG2* is almost exclusively expressed in adipose tissue [48]. However, recent studies have identified low levels of PPAR-*γ*2 expression in T cells, particularly regulatory T cells (Tregs) [49]. PPAR-*γ*Δ5 and PPAR-*γ*2Δ5 are two additional isoforms with truncated sequences resulting in non-functional LBDs. These variants are hypothesized to negatively regulate PPAR-*γ* by competing for binding sites in the DNA, without activity *per se* on gene expression. Indeed, the natural presence of PPAR-*γ*Δ5 isoforms impairs adipocyte precursor cells, contributing to adipose tissue dysfunction [50].

As mentioned earlier, PPAR-*γ* plays a fundamental role in the adipocyte lifecycle, being crucial for the initiation of the adipogenic differentiation program and maintaining the phenotype, integrity, and function of adipocytes [51,52]. Indeed, *Pparg* knockout mice showed a reduction in brown adipocytes and diminished lipid droplets compared to wild-type animals [53,54,55]. Furthermore, patients with familial partial lipodystrophy subtype 3 (FPLD3), a rare autosomal dominant condition resulting from loss-of-function mutations in the *PPARG* gene, display a deficiency of subcutaneous adipose tissue in specific regions [56,57]. Besides this adipogenic activity, PPAR-*γ* modulates lipid and glucose metabolism, as it controls the expression of genes participating in the release (e.g., lipoprotein lipase, *LPL*), transport (e.g., *CD36*), and storage of FAs, as well as gluconeogenesis (e.g., *FABP4*) [58,59].

PPAR-*γ* directly regulates the expression of genes involved in lipid transport and metabolism in myeloid cells, with direct implications in their differentiation and polarization. For instance, *PPARG* expression is highly induced during monocyte differentiation into macrophages [60,61,62] or dendritic cells [63,64,65]. Additionally, PPAR-*γ* activation drives the acquisition of an alternative M2 macrophage phenotype [66,67,68], which could result in enhanced metastasis in lung [69,70] and prostate [71] cancers. PPAR-*γ* is also crucial for the differentiation of Treg cells in visceral adipose tissue, driving a unique Treg population that controls inflammation in adipose tissue and influences insulin sensitivity [49,72].

However, except for specific contexts, the role of PPAR-*γ* when expressed or activated in cancer cells is mainly anti-tumoral, leading to a reduction in cell growth or invasiveness. For example, administration of the synthetic ligand troglitazone inhibited prostate cancer cell proliferation in vitro and in vivo [73]. Similarly, PPAR-*γ* activation reduced proliferation through estrogen receptor signaling inhibition [74]. In thyroid cancer mouse models, PPAR-*γ* activation via rosiglitazone decreased cell proliferation and delayed tumor progression by inducing apoptosis [75]. Interestingly, treatment with rosiglitazone repressed tumor metastatic potential in gastric cancer [76], inhibited EMT and metastasis by antagonizing TGF-β signaling in lung cancer [77], and suppressed metastasis in hepatocellular carcinoma by regulating metalloproteinases (MMPs) and E-cadherin [78]. Moreover, treatment with PPAR-*γ* agonists increases sensitivity to chemotherapy in various types of cancer, including lung [79,80,81], pancreas [5], and breast cancer [82]. Although these anti-tumoral effects highlight PPAR-*γ* activation as a potential therapeutic option for diverse cancers, a consistent protumorigenic role for PPAR-*γ* has been suggested in bladder cancer [83,84,85], indicating a tissue-specific role for this transcription factor.

## 2. Cancer Heterogeneity and Cancer Stem Cells

Next-generation sequencing technologies have allowed us to recognize cancer as independent and unique entities across tumor types (intertumor heterogeneity) and patients (intratumor heterogeneity). Nowadays, it is well acknowledged that intratumor heterogeneity is mainly responsible for therapeutic failure and recurrence [86,87,88,89], thus becoming a major priority in cancer biology research. Indeed, three main causes explain the underlying mechanisms of such heterogeneity: genetic driver mutations, hierarchical organization of tumors, and the influence of the tumor microenvironment (TME). While their nature and mechanisms are divergent, these three factors are not mutually exclusive [90].

First proposed by Peter Nowell in 1976 [91], the “stochastic” clonal evolution model implies that cancer initiation, development, and progression follow Darwinian selection rules. This implies that differentiated cancer cells acquire heritable mutations, called oncogenic drivers, that confer a survival advantage to their progeny. These mutations are cumulative and beneficial for cancer cells, ultimately promoting the expansion of the fittest clones across different tumor regions [92,93,94] and contributing to intratumor heterogeneity [90].

Subsequent studies in leukemia by Tsvee Lapidot [95] demonstrated the existence of non-genetic subclonal heterogeneity in cancer, driven by the hierarchical organization of tumoral cells. The “hierarchical” cancer stem cell (CSC) model implies that intratumoral heterogeneity is sustained by a small subpopulation of cells with tumor-initiating properties undergoing symmetrical (self-renewal) and asymmetrical (differentiation) divisions. As a consequence, the cells within the tumor show differences in morphology, state of differentiation, proliferation, gene expression, metabolism, and invasive, metastatic, and angiogenic potential. CSCs have similar features to their non-tumoral counterparts, as they retain the unlimited ability of self-renewal and dedifferentiation characteristics of stem cells, though in a deregulated manner. Likewise, each subclone is an (epi-) genetically distinct entity that will lead to tumor growth in the context of such heterogeneity [96]. This early study in leukemia served as a precedent to demonstrate the CSC hypothesis in solid malignancies such as breast [97], brain [98], head and neck [99], pancreas [100,101], lung [102], prostate [103,104], colon [105,106], and sarcoma [107] cancers.

The origin of CSCs still remains unclear, and, in fact, it can vary in different cancer types. On the one hand, adult tissue resident stem cells (SCs) may undergo malignant transformation during the physiological regeneration processes that maintain tissue homeostasis [108]. On the other hand, differentiated cancer cells may acquire stemness-related properties, mediated by either EMT [109] or dedifferentiation triggered by microenvironmental signals from stromal cells [110,111]. A dual scenario in which tissue resident SCs and differentiated cancer cells originate new CSCs is present in chemoresistant pancreatic [112] and lung [113] cancer cells.

Considering all these findings, the theory explaining the origin of intratumor heterogeneity has evolved into a plastic CSC model, which implies the existence of hybrid intermediate cellular states [114]. Therefore, CSCs either self-renew or differentiate into plastic hybrid cancer cells that further differentiate or dedifferentiate according to specific signals from their niche. In this scenario, hybrid cancer cells and CSCs are continuously sensing a selective pressure that will ultimately force them to either adapt and progress or extinguish completely, thus promoting the survival of the fittest clones. For example, as malignant cells grow and the TME expands, diverse nutrients such as glucose and oxygen levels diminish in the milieu, the pH becomes acidic, and reactive oxygen species (ROS) and inflammatory mediators accumulate. Considering that differentiated tumor cells are fully glycolytic to cope with their enhanced proliferative demands (i.e., Warburg effect), this scarce scenario forces CSCs to become metabolically and functionally plastic in order to survive and detoxify their microenvironment from ROS. However, CSCs may preferentially use oxidative phosphorylation (OXPHOS) or glycolysis depending on the tumor type and experimental model used [115], and this lack of consensus could be explained, at least in part, by the intrinsic plasticity of CSCs.

Therefore, although the “hierarchical” model was controversial for more than a century [116], mounting evidence over the past decades demonstrates the presence of highly tumorigenic CSCs with self-renewal capacity and functional plasticity. CSCs bear unique features such as increased mitochondrial metabolism (i.e., OXPHOS) linked to metabolic plasticity (i.e., ability to use alternative metabolic substrates for energy production), enhanced mesenchymal-like phenotype primed to invade, increased immunoevasive properties, and an inherent ability to induce chemotherapeutic failure. For these reasons, there is an increasing interest in the study and characterization of CSCs, allowing for the design of therapeutic strategies to target this aggressive subpopulation.

## 3. PPARs in Cancer Stemness

Considering the implication of PPARs in different cancer types mentioned in the first section, and the crucial role of metabolism for CSC maintenance described above, it is not surprising that an increasing number of reports link this family of transcription factors with cancer stemness. Indeed, several studies have shown that activation of the different PPARs is higher in CSCs compared to non-CSCs, as observed in pancreatic and colorectal cancer models [117,118,119]. Importantly, a complex regulatory network involving PPAR signaling and various canonical CSC pathways modulates stemness and affects tumor progression through both metabolism-dependent and independent mechanisms. For instance, several studies have documented molecular interactions between the Wnt/β-catenin and PPAR-*γ* signaling pathways, indicating the possibility of cross-regulation occurring at various levels [120]. On the other hand, PPAR-*γ* expression is directly controlled by Notch or Sonic Hedgehog (Shh) pathways, promoting pancreatic cancer progression in murine models [121] or tumor initiation in medulloblastoma [122], respectively. Finally, direct interaction of PPAR-*δ* and the Hippo coactivator YAP1 induced *SOX9* transcription and gastric cancer progression [123].

In this section, we will review the current knowledge about the role of PPARs in cancer stemness, focusing on specific CSC features.

### 3.1. Self-Renewal and Tumor Initiation

#### 3.1.1. Introduction

Similar to their non-tumoral counterparts, CSCs express high levels of specific transcription factors, whose activity orchestrates the pluripotent state and self-renewal by regulating the expression of several other crucial genes involved in this specific cell state. For instance, NANOG, OCT3/4, and SOX2 are crucial for embryonic stem cells (ESCs) during the early stage of embryonic development [124]. Indeed, these three pluripotency factors are commonly expressed in all four types of pluripotent SCs: ESCs, adult stem cells (ASCs), induced pluripotent stem cells (iPSCs), and CSCs [124]. Additionally, some CSCs, such as pancreatic CSCs [125,126,127,128], overexpress KLF4, one of the four Yamanaka factors essential for the formation of iPSCs, together with OCT3/4, SOX2, and MYC [129].

Assessment of self-renewal ability is considered the gold standard to evaluate the presence and functionality of CSCs both in vitro and in vivo, as a surrogate marker for their tumor-initiating properties. Indeed, their resistance to anoikis and ability to form spheres when cultured as single-cell suspensions in vitro is one of the most widely used techniques to evaluate this feature. Moreover, the tumor-initiating potential of CSCs is evaluated by studying the tumor engraftment when injected in concentrations as low as a single cell (tumorigenicity assay), after initial studies by Lapidot [95].

#### 3.1.2. PPARs in Self-Renewal and Tumorigenicity

Different studies have reported that the expression and activity of PPAR-α promote the proliferation of liver [130], glioblastoma [131], and glioma [132] CSCs. In fact, knockdown of *PPARA* in human glioma stem cells not only reduced in vitro proliferation but also inhibited orthotopic xenograft tumor growth, suggesting a role in tumor initiation [132]. Moreover, incubation with specific PPAR-α agonists and knockdown assays have shown that this factor was essential for the expression of pluripotency genes and sphere-forming abilities of pancreatic and colorectal CSCs [117].

So far, the reports implicating PPAR-δ in cancer stemness are mostly restricted to intestinal cancer in animal models of HFD. Indeed, this was first reported by Beyaz et al. [40], who demonstrated that either HFD or in vivo treatment with the PPAR-δ agonist GW501516 enhanced the stemness of intestinal cells and their capacity to initiate tumors [40]. Several studies have later shown that HFD increases the self-renewal and proliferation of intestinal CSCs through PPAR-δ activation [40,133,134]. Additionally, PPAR-δ, activated by HFD, induced the expansion of colonic CSCs and induced liver metastasis by triggering the pluripotency factor *NANOG* [118].

Interestingly, and in clear contrast with the α and δ isoforms, PPAR-*γ* activity in cancer cells seems to inhibit self-renewal and pluripotency gene expression in different cancer types. Indeed, a study performed in MCF7 breast cancer cells demonstrated that activation of PPAR-*γ* or PPAR-α with pharmacological agonists showed opposite effects on the survival of mammospheres by regulating genes implicated in stemness via the NF-κB/IL-6 axis [135]: while treatment with PPAR-*γ* agonists or siRNA against *PPARA* reduced MCF7 mammosphere formation, PPAR-α agonists enhanced their number. Similarly, treatment with the PPAR-*γ* agonist 15d-PGJ_2_ significantly suppressed spheroid formation and reduced the expression of stemness-related genes in bladder cancer cells [136], as well as in glioma and glioblastoma stem cells [137,138]. Indeed, PPAR-*γ* agonists also decreased the expression of the stemness surface marker CD133 and induced glial differentiation markers in glioma CSCs. Interestingly, PPAR-*γ* agonists showed differential effects on the expression of stemness genes, as the authors observed downregulation of *SOX2* and upregulation of *NANOG* [138]. Additionally, molecular iodine reduced sphere formation and the expression of pluripotency genes and the stemness markers *CD49f* and *CK17* in cervical CSCs, through the activation of PPAR-*γ* [139].

In liver hepatoblastoma cells, a recent study demonstrated that spheroid formation was rescued by treatment with the PPAR-*γ* antagonist GW9662 in *BEX1* knockout cells, suggesting that BEX1 promoted the maintenance of hepatoblastoma stemness by inhibiting PPAR-*γ* [140]. The suggested mechanism involves metabolic regulation of cellular functions, since BEX1 promoted glycolysis to sustain self-renewal by downregulating *PPARG*/*PDK1*. Interestingly, a negative interplay between BEX1 and PPAR-*γ* has also been reported during liver regeneration. Gain- and loss-of-function experiments by pharmacological and genetic means demonstrated that PPAR-*γ* downstream BEX1 negatively regulated the expansion of liver progenitor cells, modulating *Myc* expression [141].

### 3.2. Chemoresistance

#### 3.2.1. Introduction

Overcoming resistance to chemotherapy remains a major challenge for cancer management in the pursuit of long-lasting treatment responses. While common chemotherapeutic agents are relatively efficient in eliminating bulk differentiated tumor cells, CSCs are able to escape from the treatment due to their inherent chemoresistance. Different mechanisms define this CSC feature and include, but are not limited to, slow proliferation, overexpression of ATP-binding cassette (ABC) transporters, and hyperactivation of CSC-related signaling pathways [142].

CSCs have been traditionally identified as semi-quiescent slow-proliferating cancer cells, a feature contributing to their resistance to conventional anti-proliferative chemotherapeutic drugs [143,144,145]. For example, Domenichini et al. [146] demonstrated that only slow-cycling pancreatic cancer cells capable of forming tumor spheroids in vitro exhibited resistance to therapies. Interestingly, it has been reported that colon CSCs expressed miR-215 to slow down cell proliferation, thereby evading chemotherapy-induced damage until they receive signals for proliferation and differentiation [147]. Furthermore, conventional chemotherapy is especially ineffective against CSCs when they remain dormant for extended periods [148,149].

ABC transporters are known for their ability to efflux toxic agents such as chemotherapeutic agents out of cancer cells, thereby contributing to cancer chemoresistance. Importantly, ABC transporters expression is enhanced in CSCs [150], thus representing both potential biomarkers for CSC detection and targets for anti-CSC therapy. Nevertheless, ABC transporters are not exclusively located in the cell membrane and can also be found in the membranes of cytoplasmic vesicles. For example, *ABCG2* was found to be overexpressed in CSCs from pancreatic cancer and other tumoral malignancies, where it pumps in and accumulates ABCG2-dependent substrates and drugs, such as Mitoxantrone [151].

Activation of several signaling pathways implicated in stemness, such as Hedgehog or the Notch pathways, has also been linked to CSC-intrinsic chemoresistance. For instance, the enzyme TET1, the most important demethylating enzyme implicated in chemoresistance, downregulated the Hedgehog pathway, thus favoring chemosensitivity [152]. Moreover, stromal cells mediated chemotherapy failure in pancreatic cancer by promoting the expression of the stemness regulator transcription factor *HES1* via Notch-1 receptor activation [153]. Finally, enhanced WNT/α-catenin signaling promoted increased resistance to doxorubicin in osteosarcoma by inducing the expression of the *ABCB1* transporter [154].

#### 3.2.2. PPARs in Chemoresistance

Different studies link the expression and/or activity of both PPAR-*δ* and PPAR-*γ* with stemness and chemoresistance, although their functions implicate different mechanisms that can both promote or inhibit tumoral responses to chemotherapy.

The few studies investigating the contribution of PPAR-*δ* in the acquisition of chemoresistance are concentrated on colon cancer and suggest contradictory roles for this transcription factor. On the one hand, it has been reported that PPAR-*δ* accelerated cancer cell metabolism in established cell lines via induction of *GLUT1* and *SLC1A5* expression. The authors demonstrated that this metabolic switch induced tumor progression after subcutaneous injection of colon cancer SW480 cells. Conversely, combined treatment with the PPAR-*δ* antagonist GSK0660 abrogated the enhanced tumor growth and sensitized cancer cells to cisplatin in vivo [155]. On the other hand, a study performed in primary patient-derived colorectal cancer cells demonstrated that an acidic TME promoted stemness and chemoresistance by downregulating *PPARD* expression in colorectal CSCs. Indeed, downregulation of *PPARD* reduced vitamin D receptor (*VDR*) expression and subsequently mitigated its inhibitory effect on the *SOX2* promoter, thus promoting *SOX2* expression and resulting in tumor growth and drug resistance of colorectal CSCs [156]. In fact, the authors proved that *VDR* overexpression significantly inhibited both stemness and resistance to oxaliplatin of colorectal CSCs under acidic conditions [156]. This study also presents confirmatory experiments conducted using the SW480 cell line, which yielded results consistent with those observed in primary patient-derived cells. This suggests that, most likely, environmental signals may influence the contribution of PPAR-*δ* to stemness and chemoresistance in colon cancer cells.

PPAR-*γ* can either promote or counteract chemoresistance depending on the cancer type. First, a study suggests a potential pharmacologic benefit from targeting PPAR-*γ* in chemoresistant ovarian cancer. Gene set enrichment analyses performed in samples from ovarian cancer patients indicated that PPAR signaling was increased after chemotherapy, correlating with chemoresistance. Indeed, in vitro experiments showed protection against cisplatin- and paclitaxel-induced apoptosis in ovarian cancer cell lines cocultured with human adipose tissue extracts (HATEs). Mechanistically, chemoresistance was mediated by the regulation of *ABCG2* expression by PPAR-*γ*: while activation of PPAR-*γ* with troglitazone induced *ABCG2* expression, pharmacologic targeting using GW9662 resulted in decreased *ABCG2* promoter activity and protein expression, an outcome enhanced in the presence of HATEs. Further in vivo experiments demonstrated that the combination of cisplatin with GW0662 resulted in decreased metastatic foci measured as intestinal nodules. Importantly, *PPARG* and *ABCG2* expression colocalized in human tissues from chemoresistant ovarian cancer specimens, while the expression of both proteins was reduced in the chemosensitive ones [157].

However, PPAR-*γ* has also been reported to counteract chemoresistance in lung cancer models. For instance, in non-small cell lung cancer (NSCLC), hypoxia-inducible factor alpha (HIF1α) downregulated *PPARG* expression, which, subsequently, decreased the expression of uncoupling protein 2 (*UCP2*). UCP2 disruption resulted in ROS accumulation and NRF2 stabilization, leading to *ABCG2* upregulation to enhance drug efflux. Conversely, activation of PPAR-*γ* with rosiglitazone promoted *UCP2* expression and sensitized cells to cisplatin and docetaxel [158]. Another study in lung adenocarcinoma described that *PPARG* expression was reduced in tumors resistant to targeted therapy with the EGFR inhibitor gefitinib. Indeed, the combination of gefitinib and the PPAR-*γ* agonist efatutazone resulted in increased caspase activity and decreased expression of the antiapoptotic protein *BCL2*, causing enhanced cell death. This combinatory approach also resulted in cell cycle arrest, led by *PTEN* and *P21* upregulation [159].

### 3.3. Epithelial-to-Mesenchymal Transition and Metastasis

#### 3.3.1. Introduction

The metastatic process is intimately linked to cancer stemness in different aspects. Metastasis involves a series of dynamic and reversible steps initiated by the escape of cancer cells from the primary tumor site, followed by intravasation and dissemination through the blood and/or lymphatic system, extravasation at distant organs, metastatic seeding, colonization, and tumor outgrowth in the metastatic niche. This exceptionally complex sequence of events is finely regulated by the initiation of the EMT program, which is essential in physiological conditions for the embryonic developmental process and tissue repair. This program comprises different transitory intermediate states (partial/hybrid EMT) upon activation of certain transcription factors such as Snail, Slug, Twist, and Zeb1 [160]. The cells then undergo morphological changes concomitant to increased motility and resistance to apoptosis, as well as a higher ability to degrade the extracellular matrix [161,162,163]. In this sense, the EMT program is an important contributor to tumor heterogeneity and stemness, as some studies have demonstrated that non-CSCs undergoing EMT acquire CSC-related properties during the process. This associates the EMT program with the de novo CSC emergence mentioned in the previous section [109].

However, while both non-CSCs and CSCs have the ability to undergo EMT and MET (mesenchymal-to-epithelial) processes, only CSCs have the capacity to colonize the secondary site and successfully establish a metastasis due to their inherent self-renewal and tumor-initiating properties [163]. Indeed, a significant percentage of circulating tumor cells (CTCs) express CSC markers on their surface [164,165]. For these reasons, metastasis onset can also be considered a readout of cancer stemness.

#### 3.3.2. PPARs in EMT and Metastasis

Different reports demonstrated that PPAR-α promotes EMT and metastasis onset in different cancer types. Indeed, the ketogenic enzyme HMGCS2 (3-hydroxy-3-methylglutaryl-CoA synthase) facilitates PPAR-α activation to promote *SRC* expression in colorectal and oral squamous cell carcinoma (CRC and OSCC, respectively) [166], thus inducing invasion and metastasis. Interestingly, this non-canonical role of HMGCS2 was independent of its metabolic activity, since incubation with HMGCS2 metabolites on HMGCS2-deficient cells did not restore migration nor invasion in vitro [166]. On the other hand, PPAR-α was involved in the metastatic spread of hormone-dependent cancers induced by endocrine-disrupting chemicals such as mono-(2-ethylhexyl) phthalate (MEHP). Indeed, low-dose MEHP activated the PI3K/Akt/mTOR pathway to enhance EMT-related gene expression concomitant with increased migration and invasion in vitro and metastasis in vivo in a PPAR-α-dependent manner. Interestingly, in vivo treatment with the PPAR-α antagonist GW6471 diminished the number of mesenteric metastases induced by MEHP [167].

PPAR-*δ* has been linked to metastatic dissemination in different cancers. Indeed, a comprehensive study interrogated the role of PPAR-*δ* in the metastatic process in different in vivo models across several cancer types: from tail vein experimental metastasis assays with melanoma, lung carcinoma, and colon cancer cells, to orthotopic spontaneous metastasis assays with pancreatic and breast cancer cells, as well as an intrasplenic experimental metastasis assay with colon cancer cells. In all these experiments, *PPARD* knockout cells were used to demonstrate a decrease in metastatic dissemination [34]. A very recent study by our group demonstrated that different environmental factors, such as nutrient stress or signals from stromal cells, induced ligand-dependent activation of PPAR-*δ* in pancreatic cancer, causing a profound metabolic switch [48]. Both genetic and pharmacological approaches targeting PPAR-*δ* inhibited expression of EMT-related factors, invasion in vitro, and metastasis in vivo, demonstrating its metabolism-related prometastatic activity. Moreover, PPAR-*δ* has also been shown to mediate EMT induction by stromal-derived factor 1 (SDF-1) in lung adenocarcinoma cells, via activation of CXCR4/*β*-catenin/PPAR-*δ* signaling pathway [168]. In clear contrast with the report by Zuo et al. mentioned above [34], inhibition of PPAR-*δ* with the antagonist 10 h induced the expression of mesenchymal genes (fibronectin and N-cadherin) to increase cell migration in vitro in B16/F10 mouse melanoma cells. These effects were further validated in vivo in a pulmonary metastasis assay by tail vein injection in *Ppard* knockout mice, where mice treated with 10 h showed enhanced extravasation of melanoma cells, translating into increased lung metastases [169]. Both studies utilized the same cellular model (B16/F10 cells) and in vivo metastasis induction method (tail vein injection). The differing results observed may be attributed to the distinct mechanisms of action between the small molecule drug h10, which combines ligand target and off-target effects, and the genetic approach employed by Zuo et al. [34].

While PPAR-*γ* may elicit different outcomes in terms of chemoresistance depending on the tumor type, the majority of studies on the topic indicate that PPAR-*γ* counteracts EMT, invasion, and metastasis. For instance, low *PPARG* expression correlated with clinicopathological parameters such as macroscopic vascular invasion and TNM (tumor, node, metastasis) stage in HCC patients [170]. Moreover, PPAR-*γ* activation with either rosiglitazone or troglitazone rescued the effects of TGF-β in lung cancer cells, preventing E-cadherin repression and mesenchymal markers upregulation, and thus leading to a reduced mesenchymal-like morphology. These molecular and phenotypic changes translated into decreased cell invasion in vitro and metastases in vivo, induced by TFG-β-primed cells [77]. In gastric cancer, the inhibition of PPAR-*γ* nuclear translocation and function by FABP4 has been described as an important mechanism regulating migration and metastasis. Indeed, while FABP4 deficiency resulted in increased invading cells in vitro, PPAR-*γ* activation with rosiglitazone blocked invasion and lung metastases, increasing the survival of mice injected intrasplenically with gastric cancer cells [86]. Finally, PPAR-*γ* activation with rosiglitazone downregulated the expression of metalloproteinase-2 (*MMP-2*) via *PTEN* upregulation, blocking invasion in vitro [171].

### 3.4. Immune Evasion

#### 3.4.1. Introduction

The immune system plays a dual role in tumorigenesis. On the one hand, natural immunity provides host-protecting mechanisms. On the other hand, the immune system has tumor-shaping effects, defined as the ability to promote tumor progression by escaping from immunosurveillance mechanisms [172]. Interestingly, the strong correlation found between CSC marker expression and the infiltration of immunosuppressive cells such as M2-polarized protumoral macrophages (TAMs), myeloid-derived suppressor cells (MDSCs), and regulatory T cells (Tregs) has provided clear insights into the importance of CSCs in the regulation of the tumoral immune landscape [173,174].

For example, pancreatic CSCs exhibit upregulation of specific surface markers linked to immune evasion, such as the effector T cell inhibitor programmed cell death ligand-1 (PD-L1), the antiphagocytic signal CD47, or the more recently described Peptidoglycan Recognition Protein 1, all of which were correlated with poor prognosis [127,175,176]. Additionally, secreted factors and cytokines found in the immunotolerant microenvironment, such as interferons, are also involved in the maintenance of CSC properties. On the one hand, an early study demonstrated that IFN-γ enhanced the expression of CSC markers both in vitro and in vivo while promoting invasiveness and metastasis [177]. On the other hand, a more recent study further reinforced the notion that IFNs modulate pancreatic CSC functionality. In this study, pancreatic CSC-secreted IFN-γ promoted IFN-stimulated gene 15 (ISG15) secretion by TAMs, which in turn fostered CSC maintenance and activity [178].

#### 3.4.2. PPARs and Immune Evasion

A link between PPAR-*δ* and immunoevasion has been recently reported in pancreatic cancer. On the one hand, PPAR-*δ* activation with a chemical agonist or with HFD significantly accelerated PDAC progression from PanINs via CCL2-mediated immunosuppression. PPAR-*δ* mediated the secretion of the chemokine CCL2 in the early stages of preneoplastic lesions in the pancreas, which induced the recruitment of protumoral macrophages and MDSCs, and exclusion of CD8^+^ T cells [45]. In addition to the above, an elegant study demonstrated a non-canonical function of the mitochondrial glutamic-oxaloacetic transaminase 2 (GOT2) in pancreatic cancer progression. Mechanistically, activation of PPAR-*δ* via GOT2 inhibited T cell effector recruitment in the TME. This promoted an immunoevasive scenario in vivo [46].

Although immunotherapy is currently approved for the treatment of muscle-invasive bladder cancer (MIBC), specific immunoevasive mechanisms driven by PPAR-*γ* can impair the outcome. Indeed, the expression of *PPARG^high^*/*RXRA^S427Y^*, a mutation that leads to PPAR-*γ* ligand-independent activation, correlates inversely with cytokine and IFN signaling, antigen processing and presentation, T cell markers, and immune checkpoints expression in MIBC patients. Moreover, *PPARG* expression in human MIBC tissues inversely correlated with CD8 infiltration, further confirming an association of PPAR-*γ* with immune evasion in bladder cancer [179].

Although not directly related to the main scope of this review, we would like to highlight that PPARs play an important role in mediating immune suppression in the TME when expressed directly in immune cells. For example, an elegant study on breast cancer demonstrated that *PPARA* expression in B cells promoted their transformation into tBregs, a unique type of regulatory B cells able to convert CD4 T cells into Tregs with prometastatic capability [180]. Another study on cervical cancer described that the accumulation of immunoevasive lipid-laden tumor-infiltrating dendritic cells (TIDCs) was mediated by PPAR-α, and a combination of the PPAR-α antagonist GW6471 with a PD-L1 blocking antibody led to decreased tumor burden and increased survival in mice [181]. Moreover, PPAR-*δ* was responsible for hepatic resident macrophage polarization towards an M2 protumoral phenotype [182]. Finally, a recent study described that PPAR-*γ* inhibition by GW9662 in MDSCs resulted in increased CD8 T cells infiltration due to reduced free fatty acid receptor 2 (FFAR2)-induced immunosuppressive activity [183].

## 4. PPARs Targeting in the Clinical Context

As described throughout this review, PPAR agonists and antagonists have been widely used in preclinical studies in vitro and in vivo in the context of cancer. The application of these agents has contributed both to elucidating the mechanisms by which PPARs function in various pathophysiological settings and to providing proof of concept for their potential therapeutic utility as repurposed drugs. Some of these compounds with agonistic activity are used in the management of conditions including type 2 diabetes, dyslipidemia, and metabolic syndrome. However, their clinical application has been limited by toxic or systemic side effects in several cases, and, consequently, safety considerations have moderated the enthusiasm for adapting these drugs for oncology applications.

One of the most well-known classes of PPAR agonists is the TZDs, i.e., pioglitazone or rosiglitazone, initially used for the treatment of several metabolic diseases. Supported by early preclinical data, several clinical trials have explored the role of these compounds for cancer treatment (Table 1). For instance, efatutazone has been tested as a single treatment or in combination with other chemotherapeutic agents for the treatment of metastatic and advanced solid tumors, with promising results for thyroid cancer [184]. On the other hand, pioglitazone has been tested in small, single-arm, or exploratory phase II trials in oral premalignant lesions, thyroid carcinoma, non-small cell lung cancer, and pancreatic cancer. The most consistent clinical activity was reported in chronic myeloid leukemia (CML), where the addition of pioglitazone to imatinib enhanced the depth of molecular responses in residual disease, as shown in the NCT02730195 trial. However, the U.S. Food and Drug Administration raised a warning due to an increased risk of developing bladder cancer in patients treated with pioglitazone, and it was withdrawn from the French market [185].

Indeed, certain TZDs have shown significant systemic toxicity, and some of their adverse effects were substantial enough to limit or even lead to the withdrawal of certain agents from the market. Among their most important side effects is fluid retention, which is strongly associated with an increased risk of new or worsening heart failure [186,187,188]. Early clinical trials with rosiglitazone reported a potential increase in ischemic cardiovascular events, such as myocardial infarction and heart failure [189]. Additional safety concerns included weight gain and bone fractures [190]. Similarly, another TZD, troglitazone, was withdrawn from the market after a growing number of case reports and epidemiological studies linked it to severe liver toxicity and acute liver failure [191,192].

Considering the toxic side effects associated with TZDs, research is underway to develop new PPAR-targeting compounds with different mechanisms of action and improved safety. However, concerns remain; indeed, a Phase II trial of the dual PPAR-α/γ agonist saroglitazar (NCT02609048) ended early because of increased serum transaminase levels [193], and bezafibrate, a PPAR-α/δ/γ agonist, has been linked to higher serum creatinine, indicating possible renal risks [194].

On the other hand, there is an increasing interest in the development and evaluation of molecules with inhibitory effects on the PPAR signaling pathway. For instance, the compound FX-909 is a covalent inverse PPAR-*γ* agonist [195] currently tested in a first-in-human, multicenter, open-label Phase 1 study (NCT05929235) designed to assess its safety and tolerability, and preliminary clinical activity of FX-909 given orally to patients with advanced solid malignancies, including urothelial carcinoma [196]. Moreover, the PPAR-α antagonist TPST-1120 has entered first-in-human testing. Phase I data in advanced solid tumors demonstrated acceptable safety and preliminary anti-tumor activity, with signs of synergy in combination with PD-1 blockade (NCT03829436 [197]).

Importantly, despite preclinical evidence linking PPAR-*δ* signaling to tumor biology, no clinical trials have yet been reported that specifically evaluate PPAR-*δ* agonists or antagonists in oncology. Collectively, while experiences remain fragmented and heterogeneous, emerging results indicate that PPAR modulation may hold value both as a direct anticancer strategy and as a means of enhancing established therapeutic regimens.

## 5. Conclusions

Increasing evidence suggests a prominent role for the PPAR family of receptors in features and functions tightly linked to cancer stemness, such as self-renewal, tumor initiation, chemoresistance, and immune evasion, through the transcriptional regulation of specific target genes (Figure 2). Despite their predominant involvement in energy homeostasis in physiological conditions, only part of the effects reported in the context of cancer have been directly linked to changes in tumoral metabolism. This fact suggests the crucial (and generally disregarded) implication of factors such as promoter accessibility, availability of activating ligands and the expression and activity of transcriptional co-activators and co-repressors during oncogenic transformation, tumor progression, and differentiation state; factors that will directly influence the transcriptional cascade initiated by the PPARs and that can also explain the often contradictory roles described for PPARs in different cancer types or even within the same tumor entity. On the other hand, we cannot forget that in vivo treatment with synthetic agonists/antagonists will inevitably induce systemic consequences with the potential to influence cancer progression and/or metastatic spread. In this sense, effects previously frequently attributed to PPARs modulation in tumor cells may have been overestimated. Systemic side effects of PPAR-targeting molecules also represent a great challenge for repurposing drugs for cancer treatment.

In conclusion, PPARs influence different aspects of cancer progression, including cancer stemness, through context-dependent transcriptional regulation. However, we are still very far from understanding how different cellular factors and states regulate their activity and thus affect their downstream transcriptional program. Nevertheless, targeting PPARs presents significant potential in oncology, particularly through the development of novel drugs with enhanced selectivity and safety profiles, as well as combination therapies that maximize therapeutic benefits.

## Figures and Tables

**Figure 1 cells-14-01610-f001:**
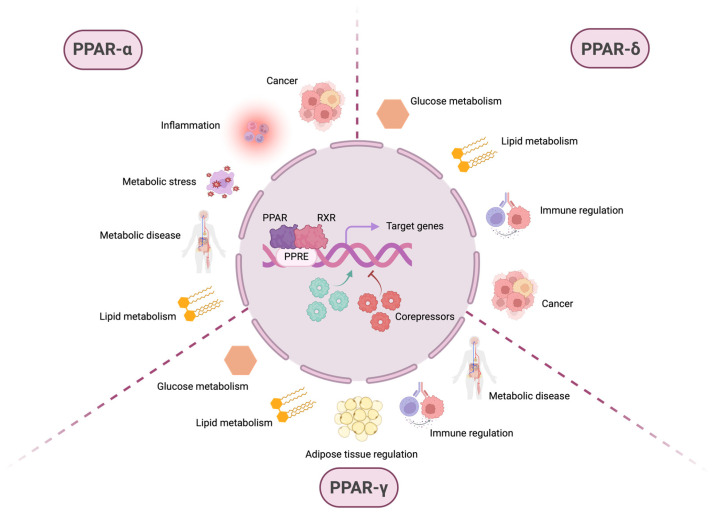
PPARs are involved in various physiological and pathophysiological processes. After ligand-dependent activation and recruitment of retinoic X receptor (RXR), the PPAR:RXR heterodimer binds to PPAR response elements (PPREs) located in the promoter regions of the DNA of target genes. The transcriptional activation or repression of these genes depends not only on this binding but also on the subsequent recruitment of coactivator or corepressor molecules, which modulate gene expression in response to different cellular contexts. Although PPARs are primarily known for their role in metabolism-related pathways, such as glucose and lipid metabolism, they also regulate inflammation, immune homeostasis, and pathological conditions, including metabolic diseases and cancer. While some of these processes involve overlapping functions among different PPAR isoforms, others are specific to individual PPARs. PPAR-α is mainly associated with lipid metabolism, but also plays a role in metabolic stress, metabolic diseases, inflammation, and cancer. PPAR-δ regulates both glucose and lipid metabolism, contributes to immune system modulation, and exhibits protumorigenic activity. PPAR-γ is involved in glucose and lipid metabolism, adipose tissue regulation, immune homeostasis, and metabolic disease. These observations highlight the broad spectrum of biological processes in which members of the PPAR family—primarily recognized as key regulators of cellular metabolism—are implicated.

**Figure 2 cells-14-01610-f002:**
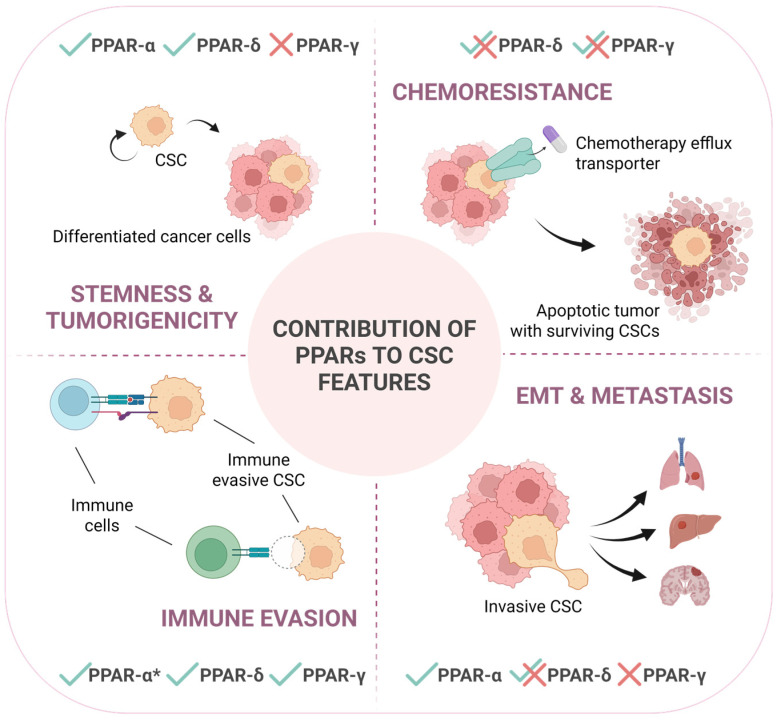
PPARs contribute to different CSC-intrinsic features in a context-dependent manner. This schematic illustrates the differential involvement of PPARs in regulating key characteristics of cancer stem cells: stemness and tumorigenicity, chemoresistance, immune evasion and epithelial-to-mesenchymal transition (EMT), and metastasis. Checkmarks (✓) and crosses (✗) indicate the positive or negative, respectively, functional contribution by each PPAR isoform to the respective process, while overlapping checkmarks with crosses (✓✗) indicate that the specific isoform is reported to impact in both manners depending on the tumor type or specific circumstances. PPAR-*α**: contribution to immune evasion of this isoform is related to its direct expression in immune cells, not in cancer cells. This underscores the context-dependent roles of PPARs in modulating CSC behavior and their potential impact on cancer progression and therapeutic failure.

**Table 1 cells-14-01610-t001:** Overview of clinical studies involving drugs that target PPARs.

Target	Drug	Mechanism of Action	NCT_ID	Indication	Phase
PPAR-α	Fenofibrate	Agonist	NCT01965834	Multiple Myeloma	Phase II
			NCT05813145	Breast Cancer	Interventional
			NCT00357500	Pediatric/Relapsed Cancers	Phase I/II
	TPST-1120	Inhibitor	NCT03829436	Advanced Solid Tumors	Phase I/1b
PPAR-γ	FX-909	Inverse agonist	NCT05929235	Solid Tumors, Advanced Urothelial Carcinoma	Phase I
	Efatutazone (CS-7017)	Agonist	NCT00408434	Advanced or Metastatic Cancer	Phase I
			NCT00967616	Colorectal Cancer	Phase II
			NCT01199068	Non-small Cell Lung Cancer	Phase Ib
			NCT02152137	Thyroid Cancer	Phase II
			NCT02249949	Mixed Liposarcoma	Phase II
	Pioglitazone	Agonist	NCT00099021	Oral Leukoplakia (chemoprevention)	Phase II
			NCT00951379	Oral Premalignant Lesions	Phase II
			NCT01655719	Metastatic Thyroid Carcinoma	Phase II
			NCT01838317	Pancreas Cancer	Phase II
			NCT01342770	Non-small Cell Lung Cancer	Phase II
			NCT00923949	Non-small Cell Lung Cancer	Pilot
			NCT02730195	Chronic Myeloid Leukemia	Phase II
			NCT05727761	Oral Leukoplakia	Phase IIa
	Rosiglitazone	Agonist	NCT00616642	Pituitary Tumors	Phase II

## Data Availability

No new data were created or analyzed in this study. Data sharing is not applicable to this article.

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
