# Peer review of "The Emerging Role of Peroxisome Proliferator-Activated Receptors in Cancer Stemness"

_cells, 2025, doi:10.3390/cells14201610_

Round 1
Reviewer 1 Report
Comments and Suggestions for Authors
Peroxisome Proliferator-Activated Receptors (PPARs), PPAR-α, PPAR-δ, and PPAR-γ, are members of a family of nuclear hormone receptors acting as transcription factors. They control the expression of genes involved in nearly all the steps of fatty acid metabolism, and, particularly, the lipid-glucose homeostasis. To function, they need to bind the ligand, a lipid, and require a partner, the retinoid X receptors (RxR), thus forming a mature and functional heterodimer. The heterodimer complex then translocates into the nucleus, where it recognizes and binds to PPAR-responsive elements (PPERs) in the promoter regions of target genes. Malfunctioning of PPARs is associated with several diseases, including metabolic disorders, inflammation, atherosclerosis, and malignancies.
The authors of the manuscript titled "The emerging role of Peroxisome Proliferator Activated Receptors in cancer stemness" review the current knowledge of the role of the three different PPAR family members in cancer, focusing on cancer stemness.
Overall, the manuscript presents some interesting hints, and from my side, I did not detect any major flaws.
Nonetheless, before publication, a couple of amendments are required.
First: both Figures require editing because the line numbering overlaps with the figures, making it difficult to appreciate and read them.
Second: for paragraphs 3.1, 3.2, 3.3, and 3.4, I would suggest not splitting them into further sub-paragraphs, but rather merging them into a single paragraph and then shortening each of them.
Reviewer 2 Report
Comments and Suggestions for Authors
This manuscript presents a comprehensive review of the current literature on the role of Peroxisome Proliferator-Activated Receptors (PPARs) in cancer stem cell maintenance. The organization of the review is good; it covers three PPAR isoforms and their specific involvement in key properties of cancer stem cells (CSCs), including self-renewal, chemoresistance, metastasis, and immune evasion. The figures are helpful and visually summarize the complex relationships discussed in the text. My concerns and suggestions are listed below.
1. The review falls short of providing a deeper critical analysis of why these discrepancies exist. For example, when discussing the conflicting roles of in colon cancer, the authors should analyze the methodologies of the conflicting studies. Are the differences due to the use of established cell lines versus patient-derived organoids? Or perhaps differences in the specific ligands, microenvironmental conditions, or genetic backgrounds of the models? A discussion of these technical details would significantly improve the review's depth and usefulness to the scientific community.
2. Another major issue is a lack of detailed discussion of the molecular mechanisms underlying the observed effects. For example, it was mentioned that BEX1 inhibits the support of stemness. A more valuable analysis would explain the molecular cascade, detailing how BEX1 directly or indirectly affects activity and how this, in turn, influences the downstream transcriptional program that maintains stemness. Adding mechanistic insights will improve the quality of the review.
Reviewer 3 Report
Comments and Suggestions for Authors
This review manuscript provides a comprehensive synthesis of current knowledge regarding the roles of PPAR isoforms (α, δ, and γ) in cancer biology, with a specific focus on cancer stemness (CSC). The authors address how PPARs regulate key CSC-associated processes such as self-renewal, chemoresistance, EMT, metastasis, and immune evasion.
The topic is timely and relevant, as the metabolic regulation of CSCs has gained increasing attention, and PPARs are attractive targets for therapeutic intervention. The manuscript is well organized and supported by an extensive reference list. However, to fully maximize its impact, the authors should improve critical evaluation, expand mechanistic insights, and strengthen the translational outlook.
Major comments
- Critical depth and novelty: The review compiles a large body of literature, but in several places it reads as a descriptive summary rather than a critical synthesis. For example, contradictory reports on PPAR-α or PPAR-δ functions in colon and liver cancer are listed but not sufficiently analyzed. The reader would benefit from a deeper discussion of why these contradictions arise — e.g., differences in ligand type (synthetic vs endogenous), metabolic context, or immune microenvironment.
- Integration with CSC pathways: Section 3 highlights PPAR regulation of CSC properties, but the mechanistic links with canonical CSC pathways (WNT/β-catenin, Hedgehog, Notch, Hippo/YAP, PI3K–Akt–mTOR) are not well developed. Given the overlap between metabolic signaling and stemness regulation, the authors should expand on how PPARs intersect with these pathways. This would enrich the mechanistic depth of the review.
- Structure and Focus: Section 1 (general biology of PPARs) is informative but too extensive for a review focused on CSCs. It could be shortened to emphasize only those features that directly relate to cancer and stemness. The saved space could be used to expand Section 3 with more detailed discussion of CSC-specific mechanisms.; Section 4 (Conclusions) is somewhat conservative. A stronger final synthesis is recommended, outlining clear research priorities and therapeutic implications (e.g., selective PPAR modulators, combination with immunotherapy, context-specific interventions).
- Translational and clinical aspects: While the review covers preclinical models extensively, the translational perspective could be strengthened. Are there ongoing or completed clinical trials investigating PPAR ligands in cancer therapy, particularly in combination with chemotherapy or immunotherapy? Including such information would enhance the clinical relevance; The systemic effects of PPAR agonists/antagonists are briefly acknowledged, but this issue deserves more attention, since off-target metabolic consequences may complicate therapeutic strategies.
Minor comments
- Abstract: The abstract is clear but could emphasize the translational impact more explicitly — e.g., potential for PPARs as therapeutic targets in CSC-driven relapse and resistance.
- Terminology and precision: In Section 3.1.2, the contrasting effects of PPAR-α and PPAR-γ agonists on mammosphere formation are well described, but the authors should clarify whether these effects were cell line–specific or generalizable across models.
- Language and style: The manuscript is generally well written, but some sentences are long and complex. A careful language edit for conciseness and readability would improve flow.. Example: page 12, lines 545–547: “Mechanistically, FAs binding to GOT2 promoted PPAR-δ activity to inhibit T cell effector recruitment in the TME, thus promoting an immunoevasive scenario in vivo.” — this could be simplified for clarity.
- Typographical issues: Page 8, line 331: “animal models oHFD” should be corrected; Ensure uniform formatting of Greek letters (α, δ, γ) throughout text and figures.
Round 2
Reviewer 2 Report
Comments and Suggestions for Authors
The authors addressed the significant concerns. I have no further suggestions.
Reviewer 3 Report
Comments and Suggestions for Authors
The authors have carefully addressed all the changes I requested. They removed the redundant sections, clarified the cancer-related focus, and improved the structure and flow of the manuscript. The addition of clinical perspectives and critical discussion further strengthens the paper. Overall, the revised version is much clearer, more concise, and significantly improved compared to the original submission.